# Molecular mechanism of phospholipid transport at the bacterial outer membrane interface

Jiang Yeow [1], Min Luo[2,3] & Shu-Sin Chng [1,4] ✉

The outer membrane (OM) of Gram-negative bacteria is an asymmetric lipid bilayer with outer leaflet lipopolysaccharides and inner leaflet phospholipids (PLs). This unique lipid asymmetry renders the OM impermeable to external insults, including antibiotics and bile salts. To maintain this barrier, the OmpC-Mla system removes mislocalized PLs from the OM outer leaflet, and transports them to the inner membrane (IM); in the first step, the OmpC-MlaA complex transfers PLs to the periplasmic chaperone MlaC, but mechanistic details are lacking. Here, we biochemically and structurally characterize the MlaA-MlaC transient complex. We map the interaction surfaces between MlaA and MlaC in *Escherichia coli*, and show that electrostatic interactions are important for MlaC recruitment to the OM. We further demonstrate that interactions with MlaC modulate conformational states in MlaA. Finally, we solve a 2.9-Å cryo-EM structure of a disulfide-trapped OmpC-MlaA-MlaC complex in nanodiscs, reinforcing the mechanism of MlaC recruitment, and highlighting membrane thinning as a plausible strategy for directing lipids for transport. Our work offers critical insights into retrograde PL transport by the OmpC-Mla system in maintaining OM lipid asymmetry.

Gram-negative bacteria assemble two lipid bilayers in their cell envelopes - the inner membrane (IM) and the outer membrane (OM) that encapsulate the cytoplasm and periplasm, respectively. At the OM, lipopolysaccharides (LPS) pack tightly with divalent cations and occupy the outer leaflet, while phospholipids (PL) reside mostly in the inner leaflet, presenting a unique bilayer with extreme asymmetry[1–3]. The outer leaflet of LPS forms a stable layer with drastically reduced permeability, offering an effective protective barrier against the entry of toxic compounds, such as detergents and antibiotics. Proper establishment and maintenance of lipid asymmetry is thus critical for OM barrier function, which allows Gram-negative bacteria to survive in the most hostile environments, including the mammalian intestinal tract.

How OM lipid asymmetry is achieved has been extensively studied. It is well known that LPS are transported and inserted into the outer leaflet of the OM via the Lpt protein bridge machinery[4]. Other major OM components, i.e. β-barrel proteins and OM lipoproteins, are assembled into the membrane via the Bam[5] and Lol[6] pathways, respectively. It is less understood how exactly PLs are brought to the OM[7], but recent studies suggest the involvement of a redundant set of AsmA-like proteins in this process[8–10]. Evidently, balancing the levels of all these OM components during active cell growth and division would be critical to ensure OM stability, and hence lipid asymmetry. In this regard, the Tol-Pal complex has been implicated in lipid homeostasis, possibly via removal of excess PLs from the OM[11]. When coordination across these pathways is sub-optimal, PLs may become mislocalized to

[1]Department of Chemistry, Faculty of Science, National University of Singapore, Singapore 117543, Singapore. [2]Department of Biological Sciences, Faculty of Science, National University of Singapore, Singapore 117558, Singapore. [3]Center for Bioimaging Sciences, Department of Biological Sciences, National University of Singapore, Singapore 117557, Singapore. [4]Singapore Center for Environmental Life Sciences Engineering, National University of Singapore (SCELSE-NUS), Singapore 117456, Singapore. ✉e-mail: chmchngs@nus.edu.sg

the outer leaflet of the OM, forming PL bilayer patches that compromise OM lipid asymmetry and barrier function[12,13]. To cope with these defects, additional cellular mechanisms are deployed to eliminate PLs that end up aberrantly in the outer leaflet of the OM. The OM phospholipase, PldA, hydrolyzes acyl chains from these PLs[14], while the OM acyltransferase, PagP, transfers an acyl chain from outer leaflet PLs to LPS and phosphatidylglycerol (PG)[15,16]. Finally, the OmpC-Mla system, an established PL trafficking pathway, extracts PLs from the outer leaflet of the OM and shuttles them back to the IM[17,18].

The OmpC-Mla system, first identified in *Escherichia coli* based on homology to the TGD2 pathway in *Arabidopsis thaliana*, comprises the OM lipoprotein MlaA in complex with osmoporin OmpC, the periplasmic chaperone MlaC, and the IM ATP-binding cassette (ABC) transporter MlaFEDB[17–19]. Removing *ompC* or any *mla* component results in PL accumulation in the outer leaflet of the OM of *E. coli*, indicating a role in maintaining OM lipid asymmetry[17,18]. To achieve that, the OmpC-MlaA complex somehow extracts PLs from the outer leaflet of the OM, hands them over to MlaC, which in turn transfers these PLs into the IM via the MlaFEDB complex[20–22]. Many recent biochemical and structural studies have provided detailed insights into ATP-dependent PL transfer steps at the IM[19,20,23–30]. In particular, we now know that when PL-bound MlaC arrives at the IM, it can spontaneously transfer the lipid molecule to the binding cavity of MlaFEDB[21]. Even though this initial process is reversible[20,21,25], ATP binding/hydrolysis by the MlaFEDB complex ultimately catalyzes the transfer of PLs into the IM, driving overall retrograde transport from the OM[20,21]. How ATP binding/hydrolysis may be coupled to PL transport in the MlaFEDB complex can be partly inferred from recently solved structures[20,26–30], though detailed mechanistic understanding is still limited.

Lipid transfer at the OM is somewhat less understood, especially how retrograde PL movement from the outer leaflet of the OM to MlaC can be facilitated in a manner not (directly) dependent on conventional energy sources at the IM. The OM lipoprotein MlaA interacts with trimeric porins OmpC and OmpF, where OmpC appears to play a more critical role in Mla function[18]. It has been established that, unlike canonical lipoproteins, MlaA resides entirely within the OM bilayer and makes extensive contacts with porins at their subunit interfaces[31,32]. Moreover, MlaA itself forms a hydrophilic channel in the OM; it comprises a 'donut'-shaped architecture (constituted largely by five horizontal helices) and a vertical 'ridge' (formed by a 'helix-turn-helix' motif) defining the central channel. The MlaA channel may provide a plausible path for PL translocation across the OM, and may be gated by conformational movements of an adjacent β-hairpin loop motif. However, we do not yet have clear ideas of how MlaA actually sits in the OM lipid bilayer, as well as the expected conformational changes that should occur in the protein during PL transport. MlaC binds PLs with presumed high affinity[24,25,33], and has been demonstrated to interact with MlaA in vivo[24] and in vitro[23]. Yet it remains elusive how exactly MlaA interacts with MlaC to allow spontaneous, productive transfer of PLs.

Here, we elucidate a molecular picture of the transient MlaA-MlaC complex during PL transfer at the OM. Using in vivo photo- and disulfide-crosslinking, we demonstrate that MlaC interacts with MlaA directly at its periplasmic base, in a manner that juxtaposes the lipid binding cavity of MlaC with the hydrophilic channel of MlaA. In addition, mutational analyses reveal that MlaA recruits MlaC to the OM via additional electrostatic interactions between the C-terminal tail helix of MlaA and a surface charged patch on MlaC. We further establish solvent-accessibility changes in a key channel residue in MlaA that are influenced by interactions with MlaC and porins (OmpC/F), as well as functionality of MlaA itself. Finally, we solve the cryo-EM structure of a disulfide-trapped OmpC-MlaA-MlaC complex in a lipid bilayer, elucidating how MlaA recruits MlaC, and how it deforms the membrane to facilitate transport. Collectively, our work provides mechanistic insights into retrograde PL transfer at the bacterial OM critical for maintaining lipid asymmetry.

## Results

### MlaC binds at the base of the hydrophilic channel of MlaA

To receive PLs from MlaA, MlaC would have to gain access to the hydrophilic MlaA channel from the periplasmic side, albeit transiently. To map the region(s) on MlaA that contacts MlaC, we replaced ~27 residues on the periplasmic face of *E. coli* MlaA separately with *para*-L-benzoylphenylalanine (*p*Bpa)[34], and examined which positions allow photoactivatable crosslinks with MlaC. We found five positions on MlaA that demonstrated strong photoactivatable crosslinks with MlaC when substituted with *p*Bpa (Fig. 1a). Another five positions exhibited weaker crosslinks to MlaC (Supplementary Fig. 1). Interestingly, these residues are all located on the C-terminal helices (Q195-N226) of MlaA extending from the channel opening in the periplasmic face (Fig. 1b), suggesting that MlaC can be juxtaposed to receive lipids from MlaA at the base of the channel.

We have also previously established the residues on MlaC that contact MlaA via the same photocrosslinking approach[24]. To determine reciprocal contact points between the two proteins, we introduced cysteines at positions on either protein that displayed strong photoactivatable crosslinks, and monitored disulfide bond formation between these MlaA and MlaC cysteine variants in cells. We tested 24 combinations of cysteine residue-pairs from four and six positions on MlaA and MlaC, respectively. Of these pairs, 11 enabled disulfide bond formation between MlaA and MlaC, indicating that these residue positions can lie in close proximity when the two proteins interact (Fig. 2a). MlaA$^{Q195C}$ and MlaC$^{Q151C}$ did not form disulfide crosslinks with any partner cysteine variants tested. In contrast, Q205, M212 and F223 on MlaA, and K128, S172 and T175 on MlaC, each allowed disulfide bond formation with multiple tested positions on their partner protein, when replaced with cysteine (Fig. 2a, b). The strongest disulfide crosslink was MlaA$^{Q205C}$-MlaC$^{V171C}$ (Fig. 2a and Supplementary Fig. 2), enabled perhaps by sufficiently long lifetime(s) of a transient interaction mode in the complex. Interestingly, Q205 lies directly at the base of the hydrophilic channel of MlaA while V171 is positioned at the entrance of the lipid binding cavity of MlaC (Fig. 2c), suggesting that the trapped MlaA-MlaC complex may be close to or approaching an arrangement(s) that could eventually lead to productive lipid transfer.

### Electrostatic interactions are necessary for MlaC recruitment

We have shown that MlaC interacts with the C-terminal α-helices at the base of MlaA. However, there is presumably an unstructured region beyond these interacting helices (G227-E251; *E. coli* numbering) missing from the reported MlaA crystal structures[32]; yet, in the recent structural model predicted by AlphaFold (AF-P76506-F1)[35], the extreme end (N238-E251) of this 'tail' region was confidently modeled as a short α-helix (Fig. 3a). To understand if this C-terminal tail helix is also important for MlaC interaction, we made several MlaA truncation constructs, and probed for MlaC interaction. Removing residues after P232 (MlaA$^{CTD4}$) completely abolished disulfide bond formation between MlaA$^{Q205C}$ and MlaC$^{V171C}$ (Fig. 3b). These variants were consequently unable to rescue SDS/EDTA sensitivity in cells lacking both MlaA and PldA, which exhibits severe OM asymmetry defects (Supplementary Fig. 3A). In fact, deletion of the last 9 amino acids on MlaA$^{D243X}$ (MlaA$^{CTD6}$), a highly negatively charged segment on the putative terminal helix, was sufficient to disrupt disulfide crosslinks with MlaC. Similar truncations in this C-terminal tail of MlaA were also recently reported, implicating this region in MlaA function, and in vitro interaction with MlaC[36]. To test the importance of electrostatics, we also constructed two sets of Arg mutants at a single face of this helix, MlaA$^{D243R/D244R}$ (MlaA$^{2DD2R}$) and MlaA$^{D247R/E251R}$ (MlaA$^{2DE2R}$) (Fig. 3a), and found that these charge-reversed variants drastically reduced disulfide crosslinks between MlaA$^{Q205C}$ and MlaC$^{V171C}$ (Fig. 3b). Specifically, the

MlaA[2DD2R] variant was only able to partially rescue SDS/EDTA sensitivity, suggesting functional importance (Supplementary Fig. 3A). Importantly, all our different MlaA variants were still able to pull down trimeric OmpC, indicating they preserve overall MlaA structure (Supplementary Figure 4). We conclude that the negatively-charged residues on the C-terminal tail helix of MlaA are also critical for MlaC interaction.

We next asked if there are specific positively-charged regions on MlaC that interact with the C-terminal tail helix of MlaA. We identified seven clusters of positively-charged residues (Arg/Lys) all around the structure of apo-MlaC[25] (Fig. 4a), and successfully converted five of these clusters to aspartates separately. MlaC[K71D] (MlaC0) and MlaC[R143D/R147D] (MlaC3) led to partial loss of disulfide crosslinks between MlaA[Q205C] and MlaC[V171C], while MlaC[R98D/K102D] (MlaC2) and MlaC[R186C/K188D] (MlaC5) had no observable impact (Fig. 4b). However, despite preserving MlaA-MlaC interactions, MlaC[R143D/R147D] (MlaC3) and MlaC[R186C/K188D] (MlaC5) were completely unable to rescue SDS/EDTA sensitivity in the ΔpldA ΔmlaC strain (Supplementary Figure 3B), suggesting these mutations affected overall MlaC function instead. Intriguingly, we found that MlaC[K84D/R90D] (MlaC1) abolished disulfide bond formation between MlaA[Q205C] and MlaC[V171C] (Fig. 4b), and was only able to partially rescue SDS/EDTA sensitivity (Supplementary Figure 3B), akin to the levels of rescue seen for MlaA[2DD2R] (Supplementary Fig. 3A). Remarkably, combining MlaA[D243R/D244R] (MlaA[2DD2R]) with MlaC[K84D/R90D] (MlaC1) fully restored disulfide bond formation (Fig. 4c), and SDS/EDTA resistance to levels similar to wildtype (Supplementary Fig. 3C), implying that

residues D243/D244 on MlaA directly contact residues K84/R90 on MlaC during MlaA-MlaC interaction.

To obtain a possible picture of the MlaA-MlaC interface, we applied AlphaFold2 to predict the structural model of the MlaA-MlaC complex[35,37,38]. Amazingly, the model not only revealed an arrangement of MlaA and MlaC that is fully consistent with our in vivo pBpa crosslinking data, it also correctly predicted the charge-charge interactions between the D243/D244 on MlaA and K84/R90 on MlaC (Fig. 4d). Quite recently, a similar MlaA-MlaC AlphaFold2 interaction model was also predicted and used to support functional analyses of MlaC variants revealed by deep mutagenesis, even though direct impact of identified mutations on MlaA-MlaC interactions had not been demonstrated[36]. The C-terminal tail helix of MlaA is connected to the rest of the protein via a flexible linker, suggesting that this tail helix might serve as a bait to recruit apo-MlaC to the OM complex via electrostatic attraction. Importantly, once MlaC is stably bound and positioned, the hydrophilic channel in MlaA becomes continuous with the lipid binding cavity of MlaC (Fig. 4d), providing a glimpse into the route for PL transfer from the outer leaflet of the OM to MlaC.

## MlaC binding induces conformational changes in the MlaA channel

Productive MlaA-MlaC interactions would be required for PL transfer from the OmpC-MlaA complex to MlaC. We hypothesize that such interactions could elicit conformational changes in MlaA to facilitate PL transfer. To test this idea, we examined the solvent accessibilities, as

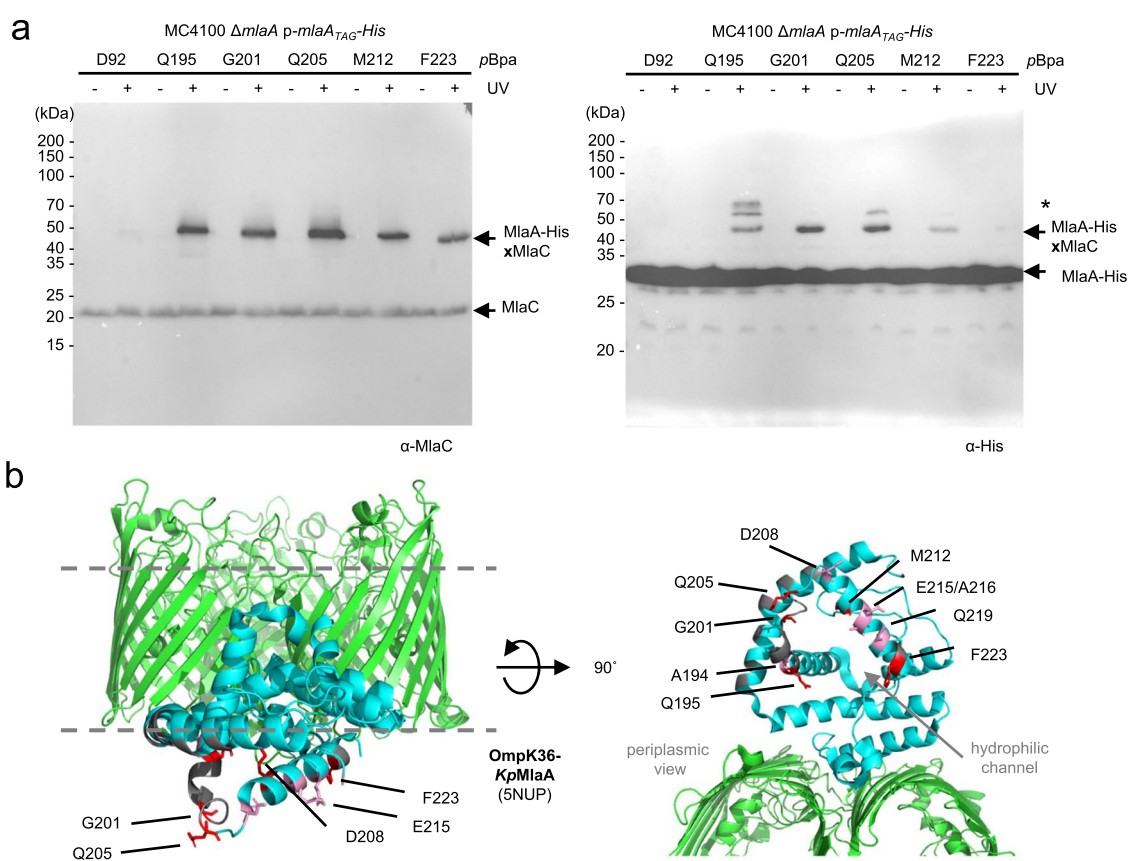

**Fig. 1 | MlaA contacts MlaC in cells via C-terminal α-helices at the periplasmic opening of its hydrophilic channel. a** Representative immunoblots showing UV-dependent formation of strong crosslinks between MlaA and MlaC in ΔmlaA cells expressing MlaA-His substituted with pBpa at indicated positions from the pCDF plasmid. Additional but unidentified MlaA[pBpa]-His crosslinked adducts were detected and denoted with an asterisk (*). The experiment had been performed at least three times with similar results. Source data are provided as a Source Data file. **b** Membrane (left) and periplasmic (right) views of cartoon representations of the crystal structure of Klebsiella pneumoniae OmpK36 (green)-MlaA (cyan) (PDB: 5NUP)[32] with positions (EcMlaA numbering) that crosslink to MlaC highlighted. Residues exhibiting strong, weak, or no photo-crosslinks with MlaC are illustrated in red and pink sticks, or in gray respectively (see Supplementary Table 4). The OM boundaries are indicated as gray dashed lines.

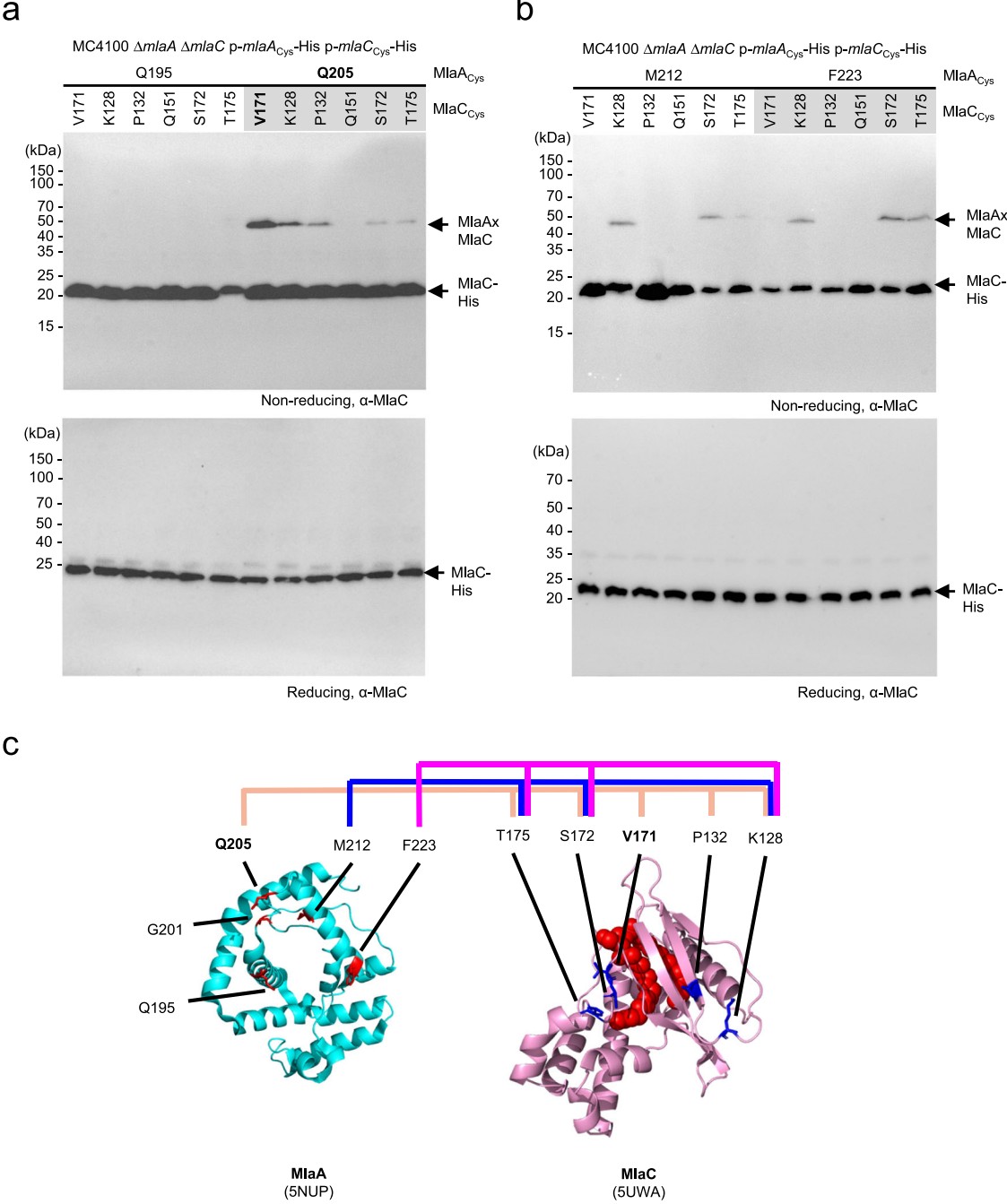

**Fig. 2 | MlaC interacts with MlaA in a manner that juxtaposes the MlaA channel and the MlaC lipid binding cavity. a, b** Representative immunoblots showing formation of disulfide crosslinks between MlaA and MlaC in Δ*mlaA* Δ*mlaC* cells expressing cysteine-substituted MlaA-His and MlaC-His from the pCDF and pET22/42 plasmids, respectively. Samples were subjected to non-reducing (*top*) or reducing (*bottom*) SDS-PAGE prior to immunoblotting. The experiment had been performed at least three times with similar results. Source data are provided as a Source Data file. **c** Cartoon representation of *Kp*MlaA (*cyan*) (PDB: 5NUP)[32] and holo-MlaC (*pink*, PL in *red* spheres) (PDB: 5UWA)[23]. Residues on MlaC which displayed disulfide crosslinking with MlaA, and vice versa, are illustrated in *red* and *blue* sticks, respectively. MlaC residues mapped to MlaA residues Q205C, M212C and F223C are connected by *beige, blue* and *pink* lines respectively.

an indication of possible conformational states, of specific residues within the MlaA channel in the absence of MlaC (Fig. 5a). We have previously shown using the substituted cysteine accessibility method (SCAM) that many hydrophilic residues in the MlaA channel are fully accessible to a membrane-impermeable thiol-reactive reagent ((2-sulfonatoethyl) methanethiosulfonate; MTSES) in wild-type cells[31]. This is true for additional channel residues identified as part of a more systematic scan for solvent accessibility (Supplementary Fig. 5). One exception is K184, which exhibited partial MTSES accessibility when substituted with cysteine; within the MlaA population in wild-type

cells, K184C is accessible to MTSES in some MlaA molecules but not others, in comparable proportions (Fig. 5b). Interestingly, removing MlaC resulted in K184C becoming MTSES-inaccessible in the entire population of MlaA (Fig. 5b), an effect not seen for other channel residues tested (Supplementary Fig. 5). This shift in accessibility of K184C was reversed by expressing wild-type MlaC in trans, but not the charge-reversed MlaC1 variant that disrupted MlaA-MlaC interactions (Fig. 5c). This suggests that interaction with MlaC is required for MlaA to adopt the state where K184C is solvent accessible. Furthermore, we observed that K184C in MlaA also became MTSES-inaccessible in the

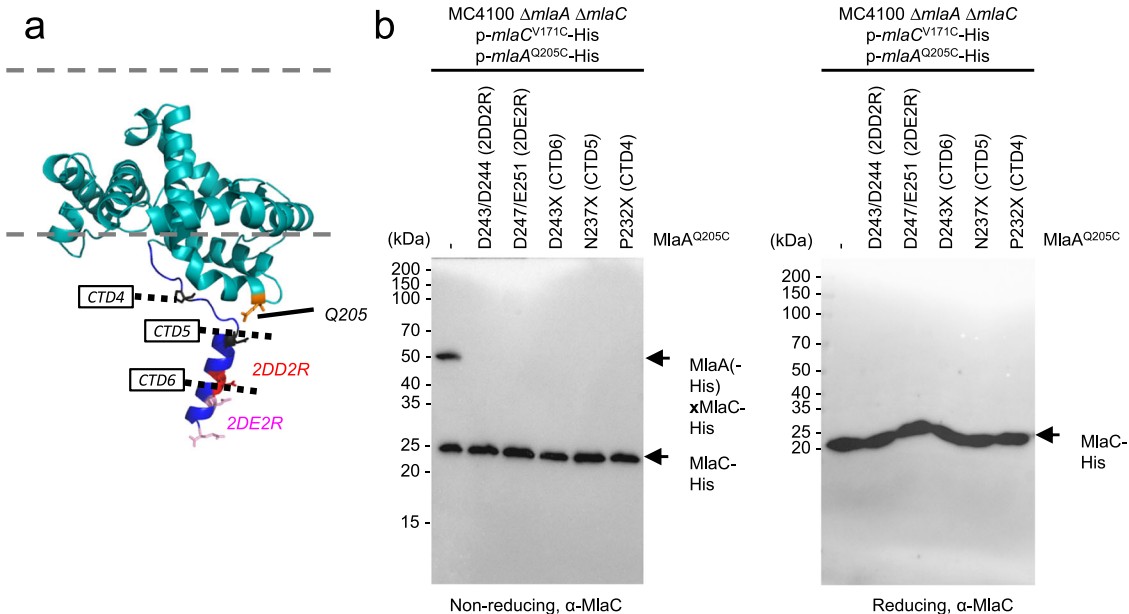

**Fig. 3 | Negatively-charged residues on the putative C-terminal tail helix of MlaA are important for MlaC recruitment. a** Cartoon representation of the *Ec*MlaA (*teal*) model generated from AlphaFold2 (AF-P76506-F1)[35] with the putative C-terminal tail helix colored *blue*. C-terminal truncation positions (CTDs) are annotated. MlaC-crosslinking residue Q205 is labeled in *orange* sticks. Charge-reversed residues in MlaA[2DD2R] and MlaA[2DE2R] are labeled in *red* and *pink* sticks respectively. **b** Representative immunoblots showing abolishment of disulfide crosslinks between MlaA and MlaC in Δ*mlaA* Δ*mlaC* cells expressing various C-terminal tail helix mutants of MlaA[Q205C](-His) and MlaC[V171C]-His from the pCDF and pET22/42 plasmids, respectively. Samples were subjected to non-reducing (*left*) or reducing (*right*) SDS-PAGE prior to immunoblotting. The experiment had been performed at least three times with similar results. Source data are provided as a Source Data file.

Δ*mlaD* strain (Fig. 5b). While MlaC is still present in this strain, it is likely bound to PLs (i.e. holo form), given that off-loading of MlaC is not possible in the absence of an intact ABC transporter in the IM. Overall, our results indicate that the binding and release of apo-MlaC may influence the solvent accessibilities and conformational states of the MlaA channel.

We have also considered the possibility that shifts in MTSES-accessibility could be due to OM asymmetry defects in these *mla* strains. This is unlikely, however, since K184C MTSES-accessibility in other OM-defective strains (Δ*tolB*, Δ*pal*)[11] was similar to that in wild-type cells (Supplementary Fig. 6A). Supporting this observation, the overexpression of phospholipase PldA, which can rescue OM asymmetry defects in *mla* mutants[17], was unable to reverse the observed shifts in accessibility of K184C in both Δ*mlaC* and Δ*mlaD* strains (Supplementary Fig. 6B). We conclude that accessibility changes in K184 in MlaA are not due to the general loss of lipid asymmetry in the OM.

### Non-functional MlaA variants no longer respond to MlaC binding

It is intriguing that K184C exhibits accessibility changes that may reflect different conformational states in the MlaA channel, possibly due to functional interactions with apo-MlaC. We next asked whether this same residue can also help us discern the functionality of MlaA itself. We first examined the MTSES-accessibility of K184C in known MlaA variants. MlaA[G141P/G143P/G145P] (MlaA[3G3P]) is a non-functional variant believed to have restricted flexibility in a β-hairpin loop that may gate the MlaA hydrophilic channel[31]. MlaA[ΔN43F44] (MlaA*) is a gain-of-function variant that is thought to have poor control of loop dynamics, resulting in PL flipping from the inner to the outer leaflet of the OM[8,31,39]. Interestingly, K184C in the entire population of MlaA[3G3P] or MlaA* appeared to be fully accessible to MTSES in cells, contrary to that observed in wild-type MlaA (Fig. 5d). Furthermore, removing MlaC or MlaD did not alter the accessibility of K184C in both MlaA variants, suggesting that MlaA[3G3P] and MlaA* are in conformational states that cannot respond to

interactions with apo-MlaC. Both MlaA[3G3P] and MlaA* are therefore non-functional in facilitating retrograde PL transfer to MlaC.

An outstanding question in the OmpC-Mla system relates to why only OmpC, but not OmpF, appears to function together with MlaA. Under high osmolarity conditions, OmpC is produced at higher levels than OmpF[18]. Therefore, to gain insights into the role of porins in the pathway without confounding effects due to expression levels, we went on to examine the solvent accessibility of MlaA K184C in the Δ*ompC* Δ*ompF* double mutant complemented in trans with either *ompC* or *ompF*. Interestingly, removing both OmpC and OmpF completely shifted the MlaA[K184C] population to a MTSES-accessible state (Fig. 5e), and expressing OmpC or OmpF in trans (at comparable levels, Supplementary Figure 7) restored the population of MlaA[K184C] to liken that in wild-type cells. These results indicate that free MlaA may not adopt a native conformational state in the OM, and that interactions with either porin is required yet sufficient for scaffolding MlaA. We further noted that the MlaA[K184C] population in just the Δ*ompC* mutant was already fully MTSES-accessible, and this observation was independent of the presence of MlaC (Fig. 5f). Overall, MlaA in strains lacking OmpC alone may also mostly not be bound to porins (owing to low levels of OmpF), and therefore largely adopts a non-functional conformational state, mirroring what we have observed in the MlaA variants. We conclude that interactions with trimeric porins allow MlaA to adopt functional states important for lipid transport.

### A molecular picture for MlaC recruitment

To directly visualize possible conformational changes in MlaA upon MlaC binding, we sought to solve the complex structure of OmpC-MlaA bound to MlaC in a native lipid environment. We exploited the strong in vivo disulfide crosslinking to trap transient interactions between MlaA[Q205C] and MlaC[V171C] and effectively isolated such OmpC-MlaA-MlaC complexes from overexpressing cells, followed by purification on size-exclusion chromatography (SEC) (Supplementary Figure 8A). Multi-angle light scattering (MALS) analysis revealed on average two MlaA[Q205C]-MlaC[V171C] bound to a trimer of OmpC in our

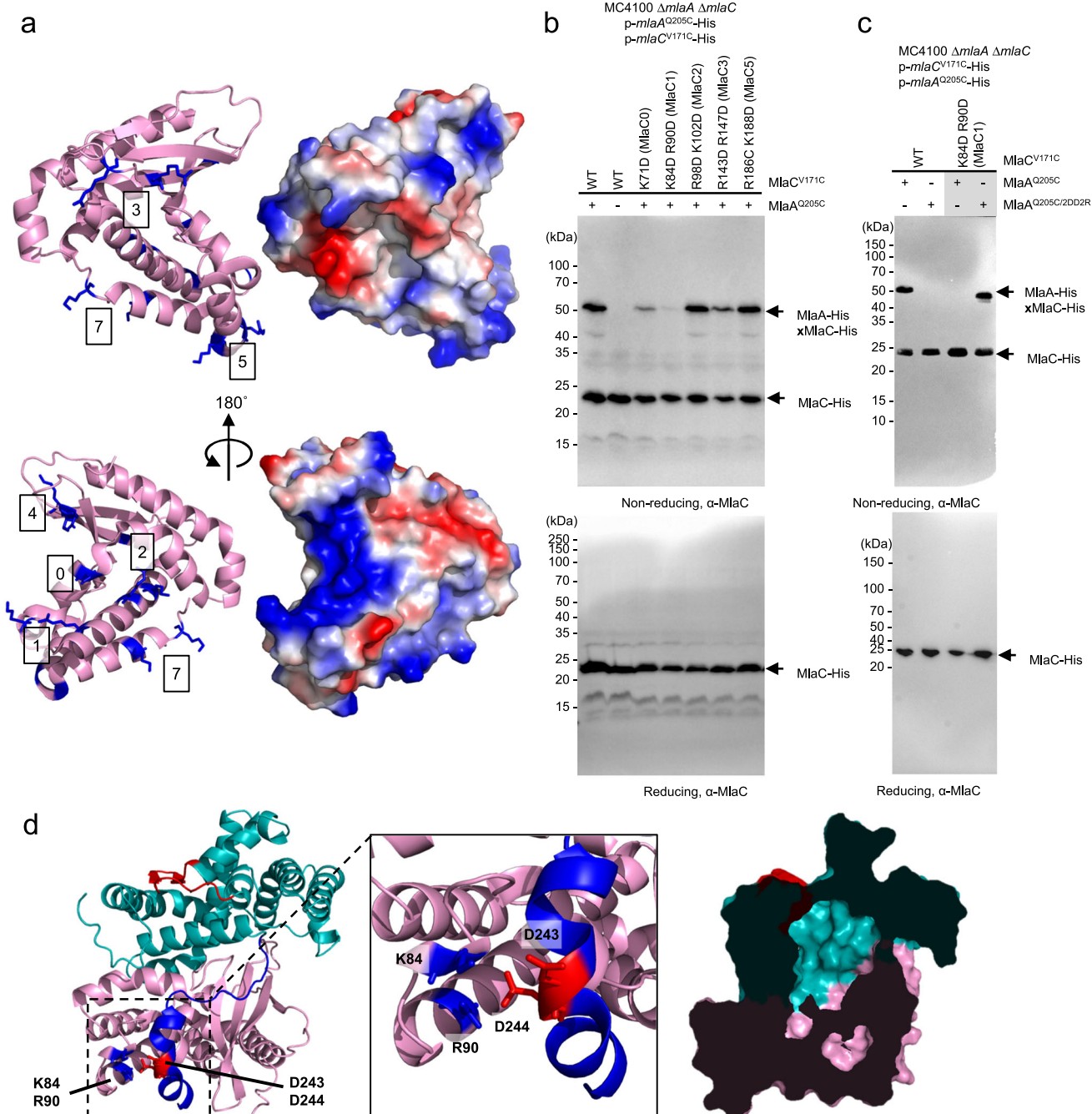

**Fig. 4 | Positively-charged surface residues on MlaC are responsible for mediating MlaA interaction. a** Cartoon and surface vacuum electrostatics representations of apo-MlaC (PDB 6GKI)[25,] revealing positively-charged patches (*blue surface*). Positively-charged residues (or groups of residues, *blue sticks*) are numbered from 0 to 7. **b** Representative immunoblots showing abolishment of disulfide crosslinks between MlaA and MlaC in ΔmlaA ΔmlaC cells expressing indicated surface charged-reversed mutants (numbered according to (**a**)) of cysteine-substituted MlaC-His and MlaA-His from pET22/42 and pCDF plasmids, respectively. The experiment had been performed at least three times with similar results. Source data are provided as a Source Data file. **c** Representative immunoblots showing restoration of disulfide crosslinks between MlaA and MlaC in ΔmlaA

ΔmlaC cells expressing charge-reversed MlaC1-His and MlaA[2DD2R]-His from pET22/42 and pCDF plasmids, respectively. Samples were subjected to non-reducing (*top*) or reducing (*bottom*) SDS-PAGE prior to immunoblotting. This experiment had been performed at least three times with similar results. Source data are provided as a Source Data file. **d** Cartoon representations of Alphafold2-predicted MlaA (*teal*)-MlaC (*pink*) model, revealing the proximity of residues D243/D244 on MlaA, and residues K84/R90 on MlaC illustrated in *blue* and *red* sticks on MlaA and MlaC respectively (inset). Cutaway surface representation of this MlaA-MlaC model is shown on the right, revealing the juxtaposition of the hydrophobic lipid-binding cavity of MlaC with the hydrophilic channel of MlaA.

purified detergent preparations (Supplementary Figure 8B). We reconstituted these OmpC-MlaA-MlaC complexes into lipid nanodiscs assembled using the MSP2N2 membrane scaffold protein[40] and *E. coli* polar lipids (Supplementary Figure 9) for single particle cryo-electron microscopy (cryo-EM) analysis.

Initial 3D reconstruction without application of symmetry operators revealed three distinct classes of OmpC-MlaA-MlaC complexes, varying in the number of MlaA-MlaC pairs bound per OmpC trimer (i.e. $OmpC_3$-(MlaA-MlaC)$_{1-3}$) (Supplementary Figure 10)[41]. However, weak map densities suggested low occupancies for the additional

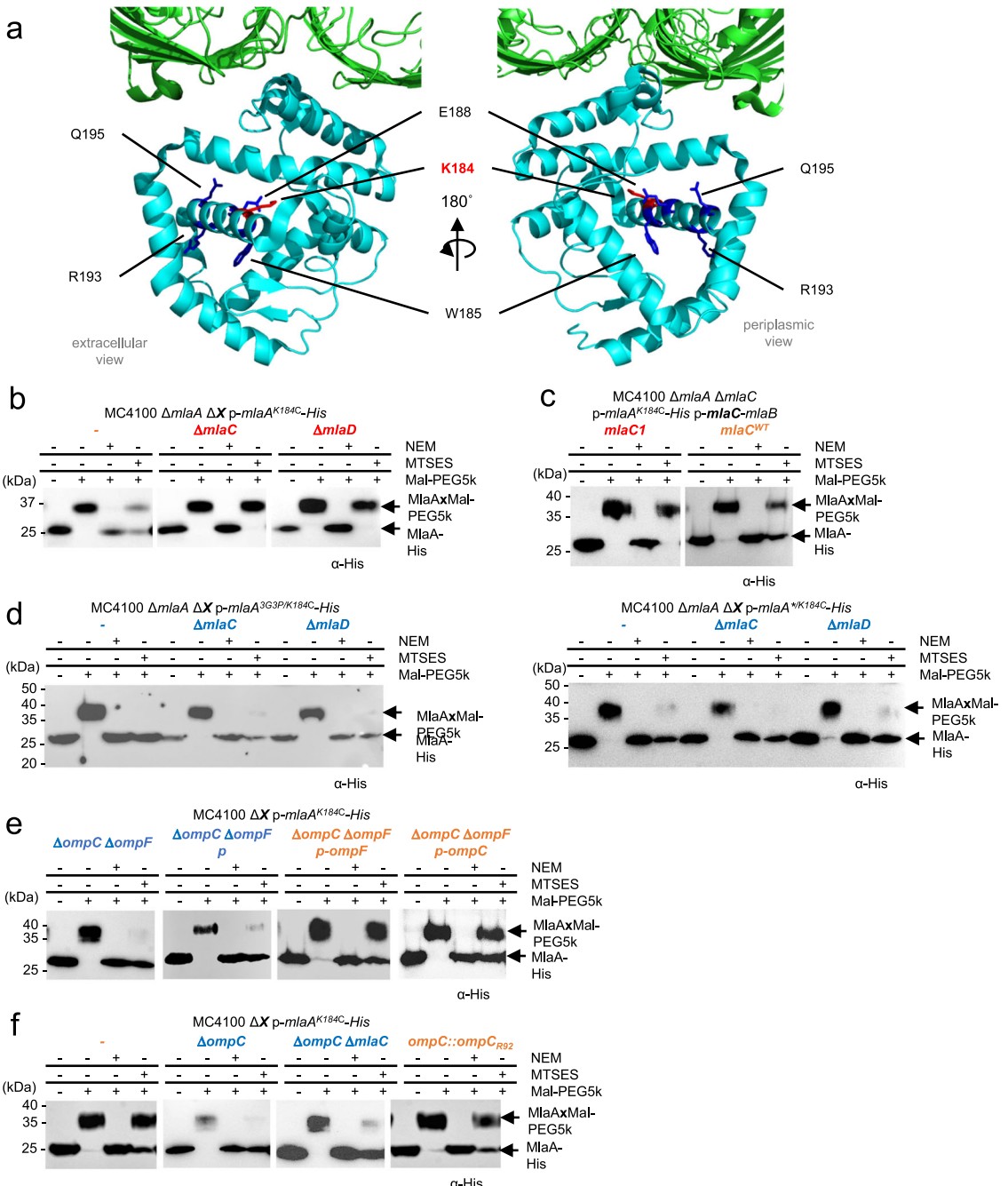

**Fig. 5 | The MlaA hydrophilic channel exhibits solvent-accessibility changes due to interactions with MlaC and trimeric porins. a** Extracellular (*left*) and periplasmic (*right*) views of cartoon representations of the reported crystal structure of OmpK36 (*green*)-*Kp*MlaA (*cyan*) (PDB: 5NUP)[32], showing channel residues substituted with cysteine (for SCAM) in *sticks*. **b, c** Representative immunoblots showing Mal-PEG5k alkylation of MlaA$^{K184C}$-His cysteine variant expressed from the pCDF plasmid, either in (**b**) Δ*mlaA*, Δ*mlaA* Δ*mlaC*, and Δ*mlaA* Δ*mlaD* backgrounds, or (**c**) Δ*mlaA* Δ*mlaC* background also producing wildtype or charge-reversed K84D/R90D (MlaC1) MlaC variants from the pET23/43-*mlaCB* plasmid. **d** Representative immunoblots showing Mal-PEG5k alkylation of MlaA$^{K184C}$-His variant also harboring non-functional 3G3P or gain-of-function *mlaA** mutations expressed from pET23/

42 plasmids in Δ*mlaA*, Δ*mlaA* Δ*mlaC*, and Δ*mlaA* Δ*mlaD* background strains. **e, f** Representative immunoblots showing Mal-PEG5k alkylation of MlaA$^{K184C}$-His cysteine variant expressed from the pCDF plasmid in various porin mutant backgrounds. In (**e**), OmpC or OmpF were expressed from pDSW206 plasmids where indicated. In SCAM, cells were labeled with membrane-permeable (NEM) or impermeable (MTSES) reagents, followed by alkylation with Mal-PEG5k, which introduces a ~5-kDa mass shift to MlaA$^{K184C}$-His. The levels of solvent accessibility of K184C in MlaA in the various strains, i.e. fully, partially, or not blocked by MTSES, are highlighted in *blue*, *orange*, or *red*, respectively. These experiments had been performed at least three times with similar results. Source data are provided as a Source Data file.

MlaA-MlaC pairs (Supplementary Figures 10 and 11). Consequently, combining particles from these classes yielded an improved density map of OmpC$_3$-(MlaA-MlaC) with an overall resolution of 2.9 Å (Supplementary Figure 10A, C).

The membrane-embedded components of OmpC$_3$-MlaA were well-resolved with side chains clearly visualized, while the density of

bound MlaC was limited to 5.0-7.0 Å (Fig. 6a) owing to continuous flexibility[42]. The overall conformations of the OmpC trimer and MlaA in our complex (Fig. 6b) largely resembled those in reported crystal structures of *E. coli* OmpC alone (PDB: 2J1N) and MlaA in complex with porins (PDB: 5NUO, 5NUP, 5NUQ, 5NUR) (Supplementary Figure 12). While MlaA binds the OmpC trimer at the porin subunit interface in a

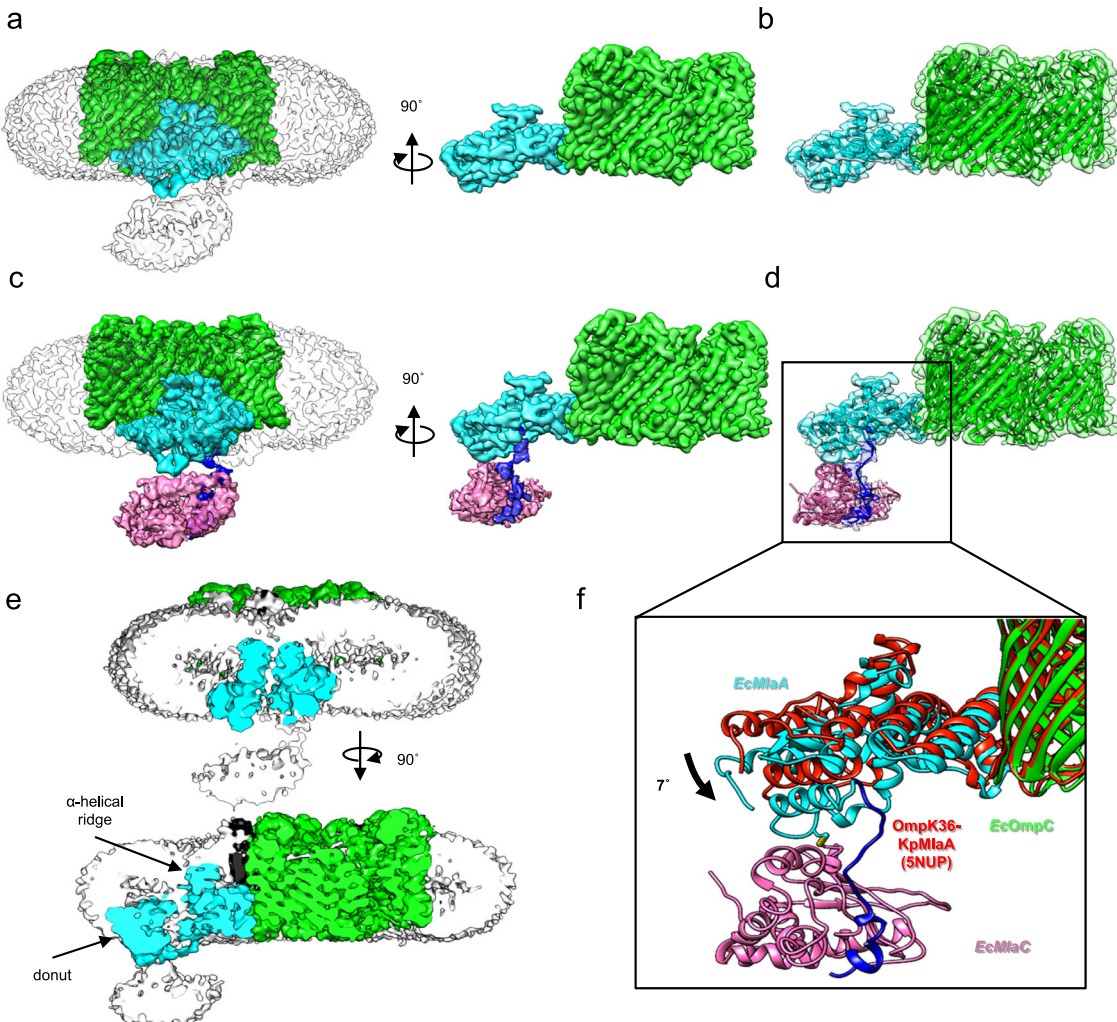

**Fig. 6 | Cryo-EM structure of OmpC$_3$-MlaA$^{Q205C}$-MlaC$^{V171C}$ in nanodiscs. a** Front and side orientations of the density map of OmpC$_3$-(MlaA-MlaC)$_{1-3}$ EMD-35250 (unsharpened; contour level of 0.06, *transparency 80%*) with the protein surface densities colored *green* (OmpC; contour level of 0.1) and *cyan* (MlaA; contour level of 0.1), respectively. **b** Cartoon illustrations of the OmpC$_3$-MlaA structure PDB: 8I8R well-fitted and refined within protein surface densities in (**a**) (*transparency 80%*). **c** Front and side orientations of the density map of OmpC$_3$-(MlaA-MlaC) EMD-35253 (unsharpened; contour level of 0.06, *transparency* 80%) with the protein surface densities colored *green* (OmpC; contour level of 0.1), *cyan* (MlaA; contour level of 0.1), *blue* (C-terminal tail helix of MlaA; contour level of 0.06), and *pink* (MlaC; contour level of 0.06), respectively. **d** Cartoon illustrations of the OmpC$_3$- MlaA-MlaC structure PDB: 8I8X well-fitted and refined within protein surface densities in (**c**) (*transparency* 80%) respectively. Two contact points between MlaA and MlaC are clearly revealed (see (**f**) inset), one modeled with the engineered MlaA$^{Q205C}$-MlaC$^{V171C}$ disulfide bond, and the other with the C-terminal linker and tail helix of MlaA. **e** Cut-away views of the density maps from (**a**) where the unsharpened nanodisc surface density is now colored *gray* (*transparency* 0%) revealed localized membrane thinning in the outer leaflet of the bilayer all around the position of the α-helical ridge of MlaA. **f** Superimposition of OmpC$_3$-MlaA-MlaC (PDB: 8I8X) and OmpK36-*Kp*MlaA (PDB: 5NUP, *red*) structures revealed a 7° tilt of MlaA towards the periplasm. Illustrations were generated using UCSF Chimera[57].

similar arrangement to that of porin-MlaA complexes[31,32], we did observe a slight downward tilt (~7 degrees) of MlaA pivoted at this contact site (PDB: 5NUP) (Fig. 6c). This causes MlaA to lean towards the 'periplasmic side' of the nanodisc, which could either be due to the presence of a lipid bilayer and/or bound MlaC. In our nanodisc structure, we can clearly visualize the MlaA donut positioned exactly at the membrane-water boundary of the inner leaflet of the bilayer, with the C-terminal part of the protein protruding into periplasmic space and connected to MlaC at two contact points (Fig. 6c). Unbiased rigid body fitting of the MlaC model alone in the density positioned V171C adjacent to Q205C (on MlaA), exactly consistent with the engineered disulfide bond forming one of the contact points (Fig. 6c). In addition, we were able to trace and fit a poly-alanine model of the C-terminal tail helix of MlaA into residual density not occupied by MlaC, accounting for the second contact point. In the final real-space refined model, the manner by which the C-terminal helix of MlaA interacts with MlaC

agrees with the AlphaFold2 prediction and strongly validates our biochemical observations (Fig. 4), solidifying the idea that this interaction is critical for MlaC recruitment. Unfortunately, it appears that we have captured MlaA-MlaC in a post-recruitment, pre-docking state, presumably constrained by the engineered disulfide linkage; that MlaC is not fully docked to the base of MlaA precluded observation of possible conformational changes expected during lipid transfer.

Regardless, our detailed molecular model of the complex, specifically MlaA, in a lipid bilayer enabled us to draw important mechanistic insights for the transfer reaction. While the MlaA donut sits in the inner leaflet of the bilayer, the channel ridge only protrudes midway into the outer leaflet. A really striking consequence of this placement is the mismatch between its membrane-spanning region and the thickness of the bilayer, leading to local membrane 'thinning', where the outer leaflet membrane-water boundary prominently bends inwards all around the position of the ridge (Fig. 6e). This deformation essentially

creates a 'funnel' that likely selects for and guides outer leaflet PLs (headgroup first) into the hydrophilic channel of MlaA. Our structure of OmpC$_3$-MlaA in a bilayer therefore reveals a possible mechanism for the initial steps leading to the removal of PLs from the outer leaflet of the OM.

## Discussion

How the OmpC-MlaA complex transfers PLs from the outer leaflet of the OM to MlaC, as part of the role of the OmpC-Mla system in maintaining lipid asymmetry, is not known. In this study, we have mapped the interacting surfaces between MlaA and MlaC, demonstrated conformational changes in MlaA during PL transfer to MlaC, and resolved the molecular architecture of an engineered OmpC$_3$-MlaA-MlaC complex in a membrane bilayer. Using molecular cross-linking and computational modeling, we have shown that MlaC docks at the periplasmic face of MlaA, possibly in a manner where the lipid binding cavity of MlaC is aligned with the hydrophilic channel of MlaA to create a continuous path for PL movement. Furthermore, we have demonstrated that electrostatic interactions between the C-terminal tail helix of MlaA and a charged patch on MlaC are required for initial recruitment. We have also uncovered that wild-type MlaA adopts distinct conformational states that is likely modulated by interactions with (apo)-MlaC. Finally, we have elucidated the cryo-EM structure of nanodisc-embedded OmpC$_3$-MlaA-MlaC trapped in a pre-docking state, revealing features that substantiated the mechanism for MlaC recruitment. While structural information on MlaA conformational changes was absent, we discerned a unique localized thinning effect on the bilayer by MlaA, pointing towards a possible way by which PLs may be funnelled into the MlaA channel. Our work provides critical mechanistic insights for the PL transfer reaction from OmpC-MlaA to MlaC for onward shuttling to the IM.

The requirement of electrostatic interactions between MlaA and MlaC allows us to infer a mechanism for how MlaA ensures effective recruitment and binding of apo- over holo-MlaC, thereby facilitating overall retrograde transfer of PLs. We have established that the positively-charged patch formed by residues K84/R90 on MlaC and the negatively-charged C-terminal tail helix residues D243/D244 on MlaA directly interact (Fig. 4), consistent with our structural observations (Fig. 6), and are important for MlaC recruitment. We note that residues K84/R90 on MlaC are exposed and accessible on apo-MlaC[25]. Remarkably, this positively-charged patch is somewhat occluded by movement of a surface loop on MlaC upon PL binding[23] (Fig. 7), limiting access by the MlaA tail helix. Such differential accessibility on MlaC likely allows preferential recruitment of apo-MlaC over holo-MlaC at the OM (Fig. 7). Post-recruitment, additional interactions between MlaA and MlaC occur at the base of the MlaA hydrophilic channel, stabilizing the complex for productive PL transfer. Once MlaC picks up a PL molecule, however, the charge-charge interaction with the MlaA tail helix is lost due to steric hindrance (Fig. 7), leading to possible destabilization of the MlaA-MlaC complex, and release of holo-MlaC from the OM. In a way, the MlaA tail helix, possibly extended almost halfway into the periplasmic space via a flexible linker (25 residues, ~88 Å), serves both as an electrostatic bait for MlaC during recruitment, and also a latch that locks down the transient MlaA-MlaC complex during PL transfer. This could be a compelling strategy to favor apo-MlaC recruitment/binding, and may ensure that PL transfer takes place in a retrograde fashion.

In the context of MlaC binding and release, we can now also introduce a model for MlaA cycling between distinct conformation states during PL transfer (Fig. 7). We have shown that MlaA exists in at least two states in wildtype strains, where K184 is somehow accessible to the aqueous environment (**State B**), or not (**State A**). Interestingly, K184 is positioned midway up the MlaA channel in the membrane, and close to the gating loop; we therefore speculate that the MlaA channel may be open in **State B**, yet closed in **State A**. In the absence of MlaC,

or with disrupted MlaA-MlaC recruitment, all of MlaA appears to become confined to **State A** (channel closed). Accordingly, apo-MlaC recruitment and binding is likely required to induce MlaA to shift to the channel-open **State B**, allowing productive PL transfer. Conceivably, holo-MlaC then no longer binds MlaA in quite the same way as apo-MlaC, possibly explaining why MlaA also adopts a channel-closed **State A** in the absence of MlaD, where all of MlaC is presumably in the holo-form. Such coordination between apo-MlaC binding and conformational changes in wild-type MlaA may favor retrograde PL transfer at the OM.

Conformational analysis using K184 accessibility also facilitated our understanding of non-functional variants/states of MlaA. We have found that free MlaA (not in complex with porins), MlaA$^{3G3P}$ and MlaA* variants all adopt conformation state(s) where K184 is fully solvent accessible and is no longer responsive to the presence of MlaC (binding) (Fig. 5). Since K184 accessibility only produces binary output, however, free MlaA and these variants may well have distinct conformations that still allow K184 to be accessible to the aqueous environment. MlaA$^{3G3P}$ is believed to be a loop-locked non-functional mutant, while MlaA* is thought to have lost control of its gating loop[31]. Our study suggests that even though MlaA* has gain-of-function possibly in allowing PLs to flip across the OM to the cell surface[8,39], this variant is in fact non-functional in the native role of MlaA. Consistent with this idea, it has recently been demonstrated that MlaA* is unable to support retrograde PL transfer in vitro[20]. We have also now demonstrated that MlaA requires trimeric porins like OmpC and OmpF for scaffolding and possibly achieving functional conformations, thus overall accounting for porin function in the Mla system[18]. Curiously, we did not observe changes in K184 accessibility in a strain expressing OmpC$^{R92A}$ (Fig. 5e), a mutation at the subunit interface of OmpC trimers (where MlaA binds) that impacts OM lipid asymmetry[31]. We speculate that the extracellular loops, proximal to R92 and overlying MlaA, at the porin subunit interface could play additional roles, perhaps in helping to enrich mislocalized PLs for MlaA to gain easy access for removal. Of note, LPS tend to be bound at these porin subunit interfacial sites, as previously reported in OmpF X-ray crystal structures[43]. We have also detected densities at the same sites in our OmpC$_3$-MlaA structure that can be confidently modeled as truncated LPS (Kdo$_2$-Lipid A) molecules (Supplementary Figure 13). Interestingly, these LPS molecules appear to block possible site-specific interactions between OmpC and MlaA in the outer leaflet of the OM, as previously revealed by photocrosslinking studies[31]. We therefore propose that in the event of perturbed OM lipid asymmetry, bound LPS might be exchanged for PLs, allowing the latter to gain easy access to the MlaA channel through the membrane thinning/funnelling effect. Removing outer leaflet PLs from porin subunit interfaces eventually restores such porin-LPS, which has been implicated recently in the formation of ordered OMP lattices on the cell surface[44]. Beyond proper scaffolding of MlaA, we believe that OmpC plays yet-to-be-ascertained functions that facilitate PL transport by MlaA.

It is truly fascinating that the manner by which MlaA sits in the bilayer causes localized membrane thinning. Such membrane perturbations/deformations have in fact been documented across protein machines that mediate the insertion and/or removal of substrates from a lipid bilayer[45–47] (Fig. 7). We hypothesize that forcing outer leaflet lipids to bend towards its ridge structure offers a mechanism to favor PLs with flexible acyl tails, yet exclude highly rigid LPS molecules from, entering the MlaA channel. Furthermore, this effect facilitates the destabilization of individual PL molecules in the outer leaflet environment by reducing packing/non-covalent interactions with neighboring lipids, thereby enabling spontaneous and affinity-driven PL transfer to MlaC. MlaA is thus a uniquely evolved OM lipoprotein that catalyzes lipid gymnastics and movement. Ultimately, the exact mechanism by which the OmpC-MlaA complex extracts mislocalized PLs from the outer leaflet of the OM and transfers them to MlaC will

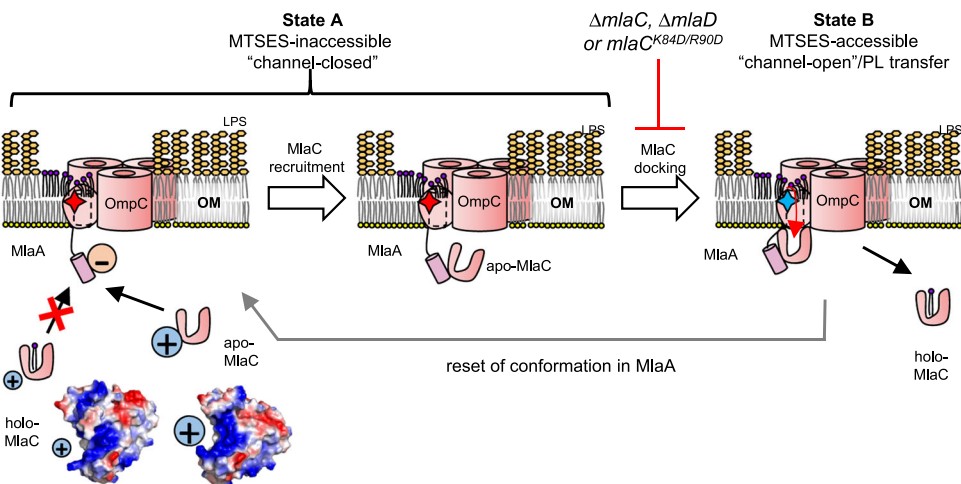

**Fig. 7 | A proposed model for MlaC recruitment/binding and accompanying MlaA conformational changes that facilitate retrograde PL transfer by the OmpC-MlaA-(MlaC) complex.** At the OM, apo-MlaC is preferentially recruited to MlaA via electrostatic interactions between MlaC surface charge patch and MlaA C-terminal tail helix. Subsequent docking of MlaC to the base of the MlaA channel induces a conformational change in MlaA, from an MTSES-inaccessible state (**State A**, "channel-closed", residue K184 represented by *red star*) to an MTSES-accessible state (**State B**, "channel-open", residue K184 represented by *blue star*). Along with the localized membrane thinning caused by MlaA, this conformational change may then facilitate the transfer of PLs from the outer leaflet of the OM into the lipid binding cavity of MlaC. Holo-MlaC then leaves the OmpC-MlaA complex to shuttle the PL ligand to the IM, thus resetting MlaA conformation. Vacuum electrostatic surface representations of MlaC bound with lipids (PDB: 5UWA)[23], and in the unliganded form (PDB: 6GKI)[25].

require additional resolution into the various conformational states we have revealed herein. Structural information of OmpC-MlaA complexes with MlaC fully docked, as well as those of mutant MlaA variants, would provide further mechanistic insights into the process. Given that OM lipid asymmetry is critical for overall barrier function, insights into the OmpC-MlaA-(MlaC) complex will also guide us in developing useful strategies to inhibit the OmpC-Mla pathway in the future.

## Methods

### Bacterial strains and plasmids
All strains, plasmids, and primers used are listed in Supplementary Table 1, 2, and 3 respectively.

### Growth conditions
Luria-Bertani (LB) media and agar were prepared at 2.5% (w/v), and 1% (w/v) respectively[48]. Unless otherwise noted, ampicillin (Amp) (Sigma–Aldrich, MO, USA) was used at a concentration of 200 μg/mL, chloramphenicol (Cam) (Alfa Aesar, Heysham, UK) at 15 μg/mL, kanamycin (Kan) (Sigma–Aldrich) at 25 μg/mL, and spectinomycin (Spec) (Sigma–Aldrich) at 50 μg/mL. For crosslinking experiments, *para*-benzoyl-L-phenylalanine (*p*Bpa; Alfa Aesar) was dissolved in 1 M NaOH at 0.25 M, and used at 0.25 mM unless otherwise mentioned.

### In vivo photoactivable crosslinking
Briefly, amber stop codon (TAG) was introduced at selected positions in p$CDFmlaA$-*His* plasmids via site directed mutagenesis using primers listed in Supplementary Table 3. For MlaA crosslinking, MC4100 with $\Delta mlaA$::*kan* background harboring p$Sup$-BpaRS-6TRN and p$CDFmlaA_{TAG}$-*His* were used[49]. An overnight 5 mL culture was grown from a single colony in LB broth supplemented with appropriate antibiotics at 37 °C. Overnight cultures were diluted 1:100 into 10 mL of the same media containing 0.25 mM *p*Bpa and grown until $OD_{600}$ reached ~1.0. Cells were normalized by optical density before pelleting and resuspended in 1 mL ice cold TBS (20 mM Tris pH 8.0, 150 mM NaCl). Samples were either used directly or irradiated with UV light at 365 nm for 20 min at 4 °C or room temperature[34]. All samples were pelleted again and finally resuspended in 200 μL of 2 X Laemmli buffer, boiled for 10 min, and centrifuged at 21,000 x *g* in a microcentrifuge

for one min at room temperature; 15 μL of each sample subjected to SDS-PAGE and immunoblot analyses.

### In vivo disulfide bond analysis
Cells harboring p$CDFmlaA_{Cys}$-*His* and pET22/42$mlaC_{Cys}$-*His* expressing MlaA$_{Cys}$-His and MlaC$_{Cys}$-His respectively with site specific cysteine substitutions was grown overnight supplemented with appropriate antibiotics in LB broth at 37 °C. Overnight cultures were diluted 1:100 into 5 mL of the same media and grown until $OD_{600}$ reached ~1.0. Cells were normalized by optical density. All samples were pelleted again and finally resuspended in 100 μL of mixed with 2 X Laemmli buffer (reducing or non-reducing), boiled for 10 min and subjected to SDS-PAGE and immunoblotting analyses using α-MlaC antibody.

### Substituted cysteine accessibility method (SCAM)
For experiments assessing the function of MlaA mutants, cells harboring pET23/42$mlaA_{Cys}$-*His* (Amp$^R$) expressing MlaA$_{Cys}$-His in various wildtype and *mla* background strains were grown overnight and supplemented with appropriate antibiotics in LB broth at 37 °C. To assess porin function, cells harboring pDSW206-*ompC/F* (Amp$^R$) and p$CDFmlaA_{Cys}$-*His* (Spec$^R$) expressing MlaA$_{Cys}$-His and native levels of porins in $\Delta ompC$ *ompF* background were grown overnight and supplemented with appropriate antibiotics in LB broth at 37 °C. Overnight cultures were diluted 1:100 into 5 mL of the same media and grown until $OD_{600}$ reached ~1.0. Cells were normalized by optical density. 1-mL cells were grown to exponential phase ($OD_{600}$ ~ 0.6), washed twice with TBS (pH 8.0), and resuspended in 480 μL of TBS. For the blocking step, four tubes containing 120 μL of cell suspension were either untreated (positive and negative control tubes added with deionized $H_2O$) or treated with 5 mM thiol-reactive reagent N-ethyl-maleimide (NEM, Thermo Scientific) or sodium (2-sulfonatoethyl) methanethiosulfonate (MTSES, Biotium). As MTSES is membrane impermeable, it is expected to react with the free cysteine in MlaA variants only when the residue near or at the membrane-water boundaries, or in a hydrophilic channel. In contrast, NEM is expected to label all MlaA cysteine variants as it is membrane permeable. Reaction with MTSES or NEM blocks the particular cysteine site from subsequently labeling by maleimide-polyethylene glycol (Mal-PEG; 5 kDa, Sigma–Aldrich). After agitation at room temperature for 1 h,

cells were washed twice with TBS, pelleted at 16,000 x *g*, and resuspended in 100 µL of lysis buffer (10 M urea, 1% SDS, 2 mM EDTA in 1 M Tris pH 6.8). Both NEM- and MTSES-blocked samples and the positive control sample were exposed to 1.2 mM Mal-PEG-5k. After agitation for another hour with protection from light, all samples were added with 120 µL of 2 X Laemmli buffer, boiled for 10 min, and centrifuged at 21,000 x *g* in a microcentrifuge for one min at room temperature; 20 µL from each sample tubes were subjected to SDS-PAGE and immunoblot analyses.

## Over-expression and purification of protein complexes

To examine interactions of MlaA mutants with OmpC, OmpC-MlaA-His protein complexes were over-expressed and purified from BL21(λDE3) Δ*ompF*::*kan* cells[31] co-transformed with either pDSW206*ompC-His* and pCDF*mlaA*, or pDSW206*ompC* and pCDF*mlaA-His*. To obtain OmpC-MlaA$^{Cys}$-MlaC$^{Cys}$-His complexes for structural analysis, OmpC, MlaA$^{Q205C}$ and MlaC$^{V171C}$-His were over-expressed and purified from BL21(λDE3) Δ*ompF*::*kan* cells[31] co-transformed with pDSW206*ompC* and pCDF*mlaC$^{Cys}$-His-mlaA$^{Cys}$*. An overnight 10-mL culture was grown from a single colony in LB broth supplemented with appropriate antibiotics at 37 °C. The cell culture was then used to inoculate a 1-L culture and grown at the same temperature until $OD_{600}$ reached ~0.6. For induction, 0.5 mM IPTG (Axil Scientific, Singapore) was added and the culture was grown for another 3 h at 37 °C. Cells were pelleted by centrifugation at 4700 x *g* for 20 min and then resuspended in 10-mL TBS containing 1 mM PMSF (Calbiochem) and 30 mM imidazole (Sigma–Aldrich). Cells were lysed with three rounds of sonication on ice (38% power, 1 s pulse on, 1 s pulse off for 3 min). Cell lysates were incubated overnight with 1% n-dodecyl β-D-maltoside (DDM, Calbiochem) at 4 °C. Cell debris was removed by centrifugation at 24,000 x *g* for 30 min at 4 °C. Subsequently, supernatant was incubated with 1 mL Ni-NTA nickel resin (QIAGEN), pre-equilibrated with 20 mL of wash buffer (TBS containing 0.05% DDM and 50 mM imidazole) in a column for 1 h at 4 °C with rocking. The mixture was allowed to drain by gravity before washing vigorously with 10 × 10 mL of wash buffer and eluted with 10 mL of elution buffer (TBS containing 0.05% DDM and 500 mM imidazole). The eluate was concentrated in an Amicon Ultra 100 kDa cut-off ultra-filtration device (Merck Millipore) by centrifugation at 4000 x *g* to ~500 µL. Proteins were further purified by size-exclusion chromatography (AKTA Pure, GE Healthcare, UK) at 4 °C on a pre-packed Superose 6 increase 10/300 GL column, using TBS containing 0.05% DDM as the eluent.

## SDS-PAGE, immunoblotting and staining

All samples subjected to SDS-PAGE were mixed 1:1 with 2X Laemmli buffer. Except for temperature titration experiments, the samples were subsequently either kept at room temperature or subjected to boiling at 100 °C for 10 min. Equal volumes of the samples were loaded onto the gels. As indicated in the figure legends, SDS-PAGE was performed using either 12% Tris.HCl gels[50] at 200 V for 45 min. After SDS-PAGE, gels were visualized by either Coomassie Blue staining, or subjected to immunoblot analysis. Immunoblot analysis was performed by transferring protein bands from the gels onto polyvinylidene fluoride (PVDF) membranes (Immun-Blot 0.2 µm, Bio-Rad, CA, USA) using semi-dry electroblotting system (Trans-Blot Turbo Transfer System, Bio-Rad). Membranes were blocked for 1 h at room temperature by 1 X casein blocking buffer (Sigma–Aldrich), washed and incubated with either primary antibodies (monoclonal α-MlaA (Abmart, Production ID: 19421-1-3/C385)[18] (1:3000), polyclonal α-MlaC (Pacific Immunology, Production ID.: 11357)[24] (1:500), or α-His antibody conjugated to the horseradish peroxidase (Penta®His HRP Conjugate, Qiagen, Lot No.: 175023485, Hilden, Germany) at 1:3000 dilution for 1–3 h at room temperature). Secondary antibody ECL™ anti-mouse IgG-HRP (from sheep) (GE Healthcare, Lot No.: 17170583), and anti-rabbit IgG-HRP (from donkey) (GE Healthcare, Lot.: 17197685) were used at 1:3000

dilution. Luminata Forte Western HRP Substrate (Merck Millipore) was used to develop the membranes, and chemiluminescence signals were visualized by *G*:Box Chemi-XT4 (Genesys version 1.4.3.0, Syngene).

## AlphaFold2 multimer modeling of MlaA-MlaC complex

To generate predicted models of the MlaA and MlaC complex, the AlphaFold2 neural-network[35] implemented in the ColabFold pipeline[38] was used. Using the mature sequences (without signal peptides) of MlaA and MlaC, default options were specified, and multiple sequence alignments produced by MMseqs2[51] were used as input for template-free structure prediction by AlphaFold2. Structural relaxation of the final protein geometry was performed using AMBER[52] to obtain five relaxed models of MlaA-MlaC complex. The best scored model was selected and illustrated in cartoon representation using PyMOL (Schrodinger, USA).

## OM permeability studies

OM sensitivity against SDS/EDTA was judged by colony-forming-unit (CFU) analyses on LB agar plates containing indicated concentrations of SDS/EDTA. Briefly, 5 mL cultures were grown (inoculated with overnight cultures at 1:100 dilution) in LB broth at 37 °C until OD600 reached ~0.4–0.6. Cells were normalized by optical density, first diluted to OD600 = 0.1 (~$10^8$ cells), and then serially diluted (ten-fold) in LB broth using 96-well microtiter plates. 1.5 µL of the diluted cultures were manually spotted onto the plates, dried, and incubated overnight at 37 °C. Plate images were visualized by *G*:Box Chemi-XT4 (Genesys version 1.4.3.0, Syngene).

## SEC-MALS analysis to determine molar masses of complexes

Prior to each SEC–MALS analysis, a preparative SEC was performed for BSA (Sigma–Aldrich) to separate monodisperse monomeric peak and to use as a quality control for the MALS detectors. In each experiment, monomeric BSA was injected before the protein of interest, and the settings (calibration constant for TREOS detector; Wyatt Technology) that gave the well-characterized molar mass of BSA (66.4 kDa) were used for the molar mass calculation of the protein of interest. SEC-purified OmpC$_3$-MlaA$^{Cys}$-MlaC$^{Cys}$-His was concentrated to 3 mg/ml and injected into Superdex 200 Increase 10/300 GL column pre-equilibrated with TBS pH 8.0 and 0.025% DDM. Light scattering and refractive index (*n*) data were collected online using miniDAWN TREOS (Wyatt Technology) and Optilab T-rEX (Wyatt Technology), respectively, and analyzed by ASTRA 6.1.5.22 software (Wyatt Technology). Protein-conjugate analysis available in ASTRA software was applied to calculate non-proteinaceous part of the complex. In this analysis, the refractive index increment *dn/dc* values (where *c* is sample concentration) of 0.143 mL/g and 0.185 mL/g were used for DDM and protein complex, respectively[53]. For BSA, UV extinction coefficient of 0.66 ml/(mg.cm) was used. For the OmpC-MlaA-MlaC-His complex, the UV extinction coefficient was calculated to be 1.62 mL/(mg.cm), based on its observed stoichiometric ratio OmpC$_3$-(MlaA$^{Cys}$-MlaC$^{Cys}$-His)$_2$.

## Reconstitution of complexes in lipid MSP2N2 nanodiscs

The reconstitution of purified OmpC$_3$-MlaA$^{Cys}$-MlaC$^{Cys}$-His into nanodiscs was adapted from published protocols[54]. Briefly, 10 mg *E. coli* polar lipid extracts (Avanti Polar Lipids) were dissolved in 1 mL chloroform and dried overnight. Then, 1 mL Tris-buffered saline (TBS) buffer (20 mM Tris HCl pH 8.0, 150 mM NaCl) containing 25 mM sodium cholate (Sigma–Aldrich) was added to the dried lipid film and vortexed and sonicated until a clear solution was obtained. The OmpC$_3$-MlaA$^{Cys}$-MlaC$^{Cys}$-His complex, the membrane scaffold protein MSP2N2-His[40], and lipids were mixed at a molar ratio of 1:2:60 in TBS and incubated, with rocking, for 1 h at 4 °C. Bio-beads SM2 resin (Bio-Rad) was subsequently added to the solution (30 mg per 1-mL reconstitution mixture) and incubated with gentle agitation for 1 h at 4 °C. Upon removal of Bio-beads, the sample was concentrated and purified

by size-exclusion chromatography on a Superose 6 Increase 10/300 GL column (GE Healthcare) on AKTA Pure. 0.5-mL fractions were collected and subjected to reducing (R) and non-reducing (NR) SDS–polyacrylamide gel electrophoresis (SDS-PAGE) and Coomassie blue staining. Fractions containing the nanodisc-embedded complexes were pooled and concentrated on a 100-kDa cut-off ultrafiltration device (Amicon Ultra, Merck Millipore).

### Cryo-EM grid preparation and data acquisition

For sample preparation, 3.0 μL of the protein sample at a concentration of 12 mg/mL was applied to glow-discharged Quantifoil holey carbon grids (1.2/1.3, 200 mesh). Grids were blotted for 3 s with 100% relative humidity and plunge-frozen in liquid ethane cooled by liquid nitrogen using a Vitrobot System (Gatan). Cryo-EM data were collected at liquid nitrogen temperature on a Titan Krios electron microscope (Thermo Fisher Scientific), equipped with a K3 Summit direct electron detector (Gatan) and GIF Quantum energy filter. All cryo-EM movies were recorded in counting mode with SerialEM4[55] with a slit width of 20 eV from the energy filter. Movies were acquired at nominal magnifications of 105k, corresponding to a calibrated pixel size of 0.834 Å on the specimen level. The total exposure time of each movie was 6 s, resulting in a total dose of 90 electrons per $Å^2$, fractionated into 50 frames. More details of electron microscopy data collection parameters are listed in Supplementary Table 5.

### Electron microscope image processing

cryoSPARC 4.0[41] was used to process the EM data according to the flowchart in Supplementary Figure 10. Dose-fractionated movies were corrected for motion using Patch Motion Correction. To obtain a sum of all frames for each movie, a dose-weighting scheme was applied, and this sum was used for all image-processing steps except for defocus determination. Defocus values of the summed images from all movie frames were calculated using patch CTF estimation without dose weighting. Particle picking was performed using the blob picker followed by the template picker. Two- and three-dimensional (2D and 3D) classifications were carried out using "2D classification", "Ab-initio Reconstruction", and "Heterogeneous Refinement". 3D refinements were conducted using "Homogeneous Refinement" and "Non-Uniform Refinement". Eventually, we obtained three cryo-EM maps representing compositional heterogeneities in OmpC3-MlaA-MlaC complexes, namely OmpC3-(MlaA-MlaC) (EMD-35253), OmpC3-(MlaA-MlaC)2 (EMD-35252) and OmpC3-(MlaA-MlaC)3 (EMD-35251). A fourth map for OmpC3-(MlaA-MlaC)1-3 (EMD-35250) was refined, where all particles from the above classes were combined. The overall resolutions for these maps were estimated based on the gold-standard criterion of Fourier shell correlation (FSC) = 0.143, while local resolution was estimated with "Local Resolution Estimation".

### Model building and refinement

An initial model of OmpC3-EcMlaA was built using the OmpK36-KpMlaA crystal structure (PDB: 5NUP) in SWISS-MODEL[56]. The model was rigid-body fitted into the highest resolution cryo-EM map (EMD-35250 - 2.93 Å) in Chimera[57], followed by density-fitting in COOT 0.9.8[58]. Refinement in RealSpace using the program of Phenix 1.20.1-4487[59] with default parameters yielded the structure for OmpC3-MlaA (PDB: 8I8R). Next, owing to poor resolution at the MlaC region, a rigid body fitting of MlaC (PDB: 5UWA) was first performed with all side-chains removed, albeit in the EMD-35253 map (3.19 Å resolution, more density at MlaC). After that, an extra density was observed at the distal groove of MlaC from the membrane, which could be clearly traced back to MlaA. A poly-alanine model corresponding to the C-terminal of MlaA was thus built in COOT 0.9.8 and refined in RealSpace/Phenix 1.20.1-4487[59], to yield coordinates for OmpC3-MlaA-MlaC (PDB: 8I8X). The final coordinates of the asymmetric units were checked using MolProbity[60]. Maps and structures shown in the figures were generated using UCSF Chimera and COOT 0.9.8. The model building and refinement statistics are shown in Supplementary Table 5.

### Reporting summary

Further information on research design is available in the Nature Portfolio Reporting Summary linked to this article.

## Data availability

Four 3D cryo-EM maps of OmpC3-(MlaA-MlaC) have been deposited in the Electron Microscopy Data Bank under accession numbers EMD-35250 (OmpC3-(MlaA-MlaC)1-3), EMD-35251 (OmpC3-(MlaA-MlaC)3), EMD-35252 (OmpC3-(MlaA-MlaC)2), EMD-35253 (OmpC3-(MlaA-MlaC)). Two atomic coordinate files have also been deposited in the Protein Data Bank under the accession numbers 8I8R (OmpC3-MlaA) and 8I8X (OmpC3-MlaA-MlaC). Source data are provided as a Source Data file. Source data are provided with this paper.

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

## Acknowledgements

J.Y. was supported by the National University of Singapore Graduate School of Integrative Sciences and Engineering Scholarship (ISEP). This work was supported by the Singapore Ministry of Health National Medical Research Council under its Open Fund Individual Research Grant (MOH-000145) (to S.S.C.). The authors would also like to acknowledge Dr Jian SHI from the Centre for Bio-Imaging Sciences (CBIS) at the National University of Singapore (NUS) for training and microscope facility management support. We also thank Dr Jianwei LI from Department of Biological Sciences (DBS), NUS for guidance with modeling and refinement techniques.

## Author contributions

J.Y. and S.-S.C. conceptualized the study, designed the experiments, and wrote the manuscript. J.Y. performed the experimental studies and single particle analyses. M.L. performed the model building. S.-S.C. supervised the study.

## Competing interests

The authors declare no competing interests.
