## [Peer Review File · Nature Communications]

Molecular mechanism of phospholipid transport at the bacterial outer membrane interfaceREVIEWER COMMENTS

Reviewer #1 (Remarks to the Author):

In this well written manuscript Yeow and co-workers have cleverly characterised the important interactions between MlaA and MlaC that are critical for the maintenance of OM lipid asymmetry. Because lipid asymmetry enables bacteria to resist various antibiotics and detergents, the insights from this work are likely to have a real impact on the future development of antimicrobials that inhibit the surface exposed Mla system.

The in vivo analyses using cysteine-based assays are exquisite, meaningful, and well controlled - a real tour-de-force. The experiments clearly show that MlaC engages with MlaA in such a way to induce conformational changes that are expected to be functionally relevant to the outer leaflet PL extraction process. The restoration of MlaA-MlaC interaction after charge reversal is quite remarkable!

The cryoEM structures also provide unexpected insight to the mechanism due to the clever use of nanodiscs. The structures show a novel membrane disruption effect that would be expected to lower the activation energy for outer leaflet PL extraction into the MlaA channel. Although the authors appear disappointed that the MlaC was captured in association with MlaA as a so-called Post-recruitment/Pre-docking stage, it is in this reviewer's opinion that this is a serendipitous but novel finding of a pathway intermediate. Perhaps future cryoEM studies using native nanodiscs that retain the bacterial OM lipids/LPS will allow the solution of additional stages of the reaction mechanism that involve the residues identified in the in vivo experiments described here.

I do not see any obvious barriers towards publication. I have only very minor comments/questions below:

- 1) Please create a table for the sup listing all of the pBpa sites tested and their outcome.
- 2) Fig 1B is *K. pneumoniae* structure. It is not clear if the residue positions and identities are labeled according to the *E.c.* or *K.n.* proteins.
- 3) Because the Cys-Cys crosslinking experiments were not exogenously catalyzed (eg using CuSO₄ or 4-DPS) the disulfides formed relied solely on the ability of periplasmic DsbA to access the introduced residues - could truly interacting interfaces have been missed due to this (false negatives)? Could the perceived differences in crosslinking be due to steric hinderance of DsbA rather than the residues not being in close proximity?

Reviewer #2 (Remarks to the Author):

The OmpC-Mla system is implicated in the maintenance of lipid asymmetry of the outer membrane of Gram-negative bacteria by mediating retrograde transport of phospholipids from the outer leaflet of the outer membrane to the inner membrane. The periplasmic chaperone MlaC shuttles phospholipids from MlaA to MlaD by forming a transient complex with either of them. In this manuscript by Yeow et al, the authors set out to reveal the mechanistic details of phospholipid transfer from the OmpC-MlaA complex to MlaC. They show that the electrostatic interactions are important for recruiting MlaC to MlaA to form a transient complex and demonstrate that the MlaC binding modulates the conformational states of MlaA. They also report the cryo-EM structure of the OmpC-MlaA-MlaC complex trapped with an engineered disulfide bond. This is a valuable work to further our understanding of the mechanism of the OmpC-Mla system.

Major points:

- 1) In the first part of the manuscript, the authors used in vivo photo- and disulfide-crosslinking and mutagenesis to map the interfaces involved in the interactions between MlaA and MlaC, which is supported by their model of the MlaA-MlaC complex predicted by AlphaFold2. The novelty of this part is reduced by a similar model of the MlaA-MlaC complex that has been reported in a recent paper (MacRea et al, 2023).
- 2) The authors report that the electrostatic interactions between the C-terminal helix of MlaA and a charged patch on MlaC is needed for disulfide crosslinks between MlaA(Q205C) and MlaC(V171C), and conclude that these electrostatic interactions are necessary for MlaC recruitment. However, their results show that the mutations that disrupt these electrostatic interactions did not affect the overall lipid transport (no SDS/EDTA sensitivity). It seems the transient complex captured by the disulfide bond between MlaA(Q205C) and MlaC(V171C) is not important for the function. Is there an alternative possibility that there are other functionally relevant transient complexes not captured by this crosslinking method?
- 3) The cryo-EM structure of the OmpC-MlaA-MlaC complex stabilized by the MlaA(Q205C) and MlaC(V171C) disulfide linkage reported in this manuscript also demonstrates that the MlaA-MlaC is in a pre-docking state, suggesting that this disulfide crosslinking does not necessarily capture the functionally relevant state.
- 4) To avoid the biased crosslinking caused by engineered cysteine sites, the authors are encouraged to try non-specific crosslinking methods (such as GraFix) to capture some endogenous transient MlaA-MlaC complexes for cryo-EM study.
- 5) The discovery of MlaC modulating the solvent accessibility of the K184C mutant of MlaA is intriguing. But unfortunately their cryo-EM structure of the OmpC-MlaA-MlaC complex is not in the conformation that can support this finding. K184 is in close proximity to negatively charged residues. Changing it to a nearly non-polar cysteine residue could have changed the solvent accessibility of itself as well as the local conformation. Direct structural evidence or other experiments are needed to support this conclusion.

Point-by-point responses to reviewer comments
(Reviewer comments in black, responses in red)

REVIEWER COMMENTS

Reviewer #1 (Remarks to the Author):

In this well written manuscript Yeow and co-workers have cleverly characterised the important interactions between MlaA and MlaC that are critical for the maintenance of OM lipid asymmetry. Because lipid asymmetry enables bacteria to resist various antibiotics and detergents, the insights from this work are likely to have a real impact on the future development of antimicrobials that inhibit the surface exposed Mla system.

The in vivo analyses using cysteine-based assays are exquisite, meaningful, and well controlled - a real tour-de-force. The experiments clearly show that MlaC engages with MlaA in such a way to induce conformational changes that are expected to be functionally relevant to the outer leaflet PL extraction process. The restoration of MlaA-MlaC interaction after charge reversal is quite remarkable!

The cryoEM structures also provide unexpected insight to the mechanism due to the clever use of nanodiscs. The structures show a novel membrane disruption effect that would be expected to lower the activation energy for outer leaflet PL extraction into the MlaA channel. Although the authors appear disappointed that the MlaC was captured in association with MlaA as a so-called Post-recruitment/Pre-docking stage, it is in this reviewer's opinion that this is a serendipitous but novel finding of a pathway intermediate. Perhaps future cryoEM studies using native nanodiscs that retain the bacterial OM lipids/LPS will allow the solution of additional stages of the reaction mechanism that involve the residues identified in the in vivo experiments described here.

We sincerely thank the reviewer for appreciating the significance of our work, and for the encouraging and supportive feedback.

I do not see any obvious barriers towards publication. I have only very minor comments/questions below:

1) Please create a table for the sup listing all of the pBpa sites tested and their outcome.

We have included such a table (new Supplementary Table 4) as requested.

2) Fig 1B is *K. pneumoniae* structure. It is not clear if the residue positions and identities are labeled according to the *E. coli* or *K. pneumoniae* proteins.

The positions are labelled according to the *E. coli* MlaA numbering. We have now clarified this point in the figure legend.

3) Because the Cys-Cys crosslinking experiments were not exogenously catalyzed (eg using CuSO₄ or 4-DPS) the disulfides formed relied solely on the ability of periplasmic DsbA to access the introduced residues - could truly interacting interfaces have been missed due to this (false negatives)? Could the perceived differences in crosslinking be due to steric hinderance of DsbA rather than the residues not being in close proximity?

We agree with the reviewer that we may have missed other possible contact points between MlaA and MlaC using Cys-Cys crosslinking. We believe that disulfides between MlaA and MlaC cysteine variants can be formed either by DsbA or by air oxidation; therefore, the likelihood of observing Cys-Cys crosslinking depends on the oxidation potential of each Cys pair, the proximity and accessibility of each pair, as well as the residence time of each interaction state. While crosslinking likelihood may be low, we believe that not adding external oxidants allows us to trap interaction states that are possibly more native and relevant. In the future, we can indeed explore the use of external oxidants to capture more states for structural characterization. However, we would need to be careful about potentially trapping non-relevant states (i.e., false positives).

Reviewer #2 (Remarks to the Author):

The OmpC-Mla system is implicated in the maintenance of lipid asymmetry of the outer membrane of Gram-negative bacteria by mediating retrograde transport of phospholipids from the outer leaflet of the outer membrane to the inner membrane. The periplasmic chaperone MlaC shuttles phospholipids from MlaA to MlaD by forming a transient complex with either of them. In this manuscript by Yeow et al, the authors set out to reveal the mechanistic details of phospholipid transfer from the OmpC-MlaA complex to MlaC. They show that the electrostatic interactions are important for recruiting MlaC to MlaA to form a transient complex and demonstrate that the MlaC binding modulates the conformational states of MlaA. They also report the cryo-EM structure of the OmpC-MlaA-MlaC complex trapped with an engineered disulfide bond. This is a valuable work to further our understanding of the mechanism of the OmpC-Mla system.

We sincerely thank the reviewer for appreciating the significance of our work.

Major points:

1) In the first part of the manuscript, the authors used in vivo photo- and disulfide-crosslinking and mutagenesis to map the interfaces involved in the interactions between MlaA and MlaC, which is supported by their model of the MlaA-MlaC complex predicted by AlphaFold2. The novelty of this part is reduced by a similar model of the MlaA-MlaC complex that has been reported in a recent paper (MacRea et al, 2023).

Even though the MacRae et al paper has been published, we feel that their findings do not significantly diminish the novelty of our work. While they did identify mutations that render MlaA and MlaC non-functional, and demonstrate that the C-terminal tail of MlaA is important for MlaC interaction, they did not provide any additional experimental evidence showing that other identified MlaA/MlaC residues or regions actually physically contacted the partner protein. Much of their conclusion that these identified sites represent contact sites between MlaA and MlaC came from an AlphaFold Multimer prediction, which has not been validated experimentally. In stark contrast, we have experimentally determined the specific sites on MlaA that interact with MlaC using photo-crosslinking, defined point-to-point contacts between the two proteins using cysteine-cysteine crosslinking, established key interaction sites using reciprocal mutagenesis, as well as solved the cryo-EM structure of nanodiscs-embedded OmpC-MlaA in complex with MlaC that validated the manner by which MlaC interacts with the C-terminal tail helix of MlaA (albeit at lower resolution for this region). We have completed a huge body of biochemical, genetic and structural work that

provided many new and important findings about how MlaA and MlaC interacts, which we believe are far more significant than the MacRae et al paper.

2) The authors report that the electrostatic interactions between the C-terminal helix of MlaA and a charged patch on MlaC is needed for disulfide crosslinks between MlaA(Q205C) and MlaC(V171C), and conclude that these electrostatic interactions are necessary for MlaC recruitment. However, their results show that the mutations that disrupt these electrostatic interactions did not affect the overall lipid transport (no SDS/EDTA sensitivity). It seems the transient complex captured by the disulfide bond between MlaA(Q205C) and MlaC(V171C) is not important for the function.

We thank the reviewer for highlighting this point. We were also initially puzzled by the lack of impact on SDS/EDTA sensitivity in strains where the electrostatic interactions between MlaA and MlaC were disrupted. While these observations suggest that the electrostatic interactions are not absolutely required for OmpC-Mla function, we believe that such interactions would affect the efficiency of lipid transfer at the OM. In otherwise wildtype cells, the amount of mislocalized PLs that need to be removed from the outer leaflet of the OM by the OmpC-Mla system is expected to be small, so the hypothesized drop in efficiency due to the disruption of electrostatic interactions may not be critical. To potentially reveal the true impact of the loss of electrostatic interactions between MlaA and MlaC, we have now tested the abilities of MlaA and MlaC mutants in restoring SDS/EDTA resistance in strains lacking both PldA and OmpC-Mla function. In these strains, cells accumulate excessive amounts of PLs in the outer leaflet of the OM (JC Malinverni, 2009, Chong et al. 2015), thus requiring a highly efficient OmpC-Mla system. We showed that MlaA^{D243R/D244R} (MlaA^{2DD2R}) and MlaC^{K84D/R90D} (MlaC1) mutants only partially rescue SDS/EDTA sensitivity in *ΔpldA ΔmlaA* and *ΔpldA ΔmlaC* backgrounds respectively. In addition, co-expressing these two charge-reversed variants restored SDS/EDTA resistance, consistent with the recovery of interactions as judged by disulfide bond formation between MlaA^{Q205C} and MlaC^{V171C} (Fig. 4B). These new findings demonstrate that the electrostatic interactions between MlaA and MlaC are indeed important for function. We have included the new data in an updated Supplemental Figure 3, and revised the text in the manuscript.

2) (*cont'd*) Is there an alternative possibility that there are other functionally relevant transient complexes not captured by this crosslinking method?

3) The cryo-EM structure of the OmpC-MlaA-MlaC complex stabilized by the MlaA(Q205C) and MlaC(V171C) disulfide linkage reported in this manuscript also demonstrates that the MlaA-MlaC is in a pre-docking state, suggesting that this disulfide crosslinking does not necessarily capture the functionally relevant state.

4) To avoid the biased crosslinking caused by engineered cysteine sites, the authors are encouraged to try non-specific crosslinking methods (such as GraFix) to capture some endogenous transient MlaA-MlaC complexes for cryo-EM study.

We thank the reviewer for raising these related points. We agree that disulfide crosslinking may be biased to the engineered sites and does not necessarily allow us to capture the fully docked MlaA-MlaC complex, i.e. the state(s) functionally relevant for lipid transfer. Even so, we have used disulfide bond formation in cells to capture MlaA-MlaC in a pre-docking state, which revealed structural evidence for interactions between the C-terminal tail helix of MlaA with MlaC, in agreement with its functional importance. Nonetheless, we agree with the reviewer that we could explore the use of non-specific crosslinkers such as GraFix to capture

more transient states (functionally relevant for lipid transfer) in vitro for structural characterization. However, these experiments would be beyond the scope of this manuscript.

5) The discovery of MlaC modulating the solvent accessibility of the K184C mutant of MlaA is intriguing. But unfortunately their cryo-EM structure of the OmpC-MlaA-MlaC complex is not in the conformation that can support this finding. K184 is in close proximity to negatively charged residues. Changing it to a nearly non-polar cysteine residue could have changed the solvent accessibility of itself as well as the local conformation. Direct structural evidence or other experiments are needed to support this conclusion.

We agree with the reviewer that there could be potential changes to the solvent accessibility and the local conformation in our K184C mutant. However, we have previously found this variant to be fully functional (Yeow et al. 2018), indicating that it is logical to assume MlaA^{K184C} behaves like wildtype MlaA in most, if not all, aspects related to function, including conformational changes. Therefore, we believe it is reasonable to infer conformational changes in MlaA from changes in the solvent accessibility observed for K184C.

It is true that our cryo-EM structure did not reveal any conformational changes in the MlaA channel. However, the absence of observable conformational changes was likely due to the fact that MlaC was captured in a pre-docking state, but not in a state where MlaC is fully docked and engaging MlaA for lipid transfer. As such, we believe that conformational changes revealed by K184C accessibility are still relevant, but as pointed out correctly by the reviewer, additional structural evidence to capture such a state is needed in the future.

REVIEWERS' COMMENTS

Reviewer #1 (Remarks to the Author):

Yeow and co-workers, have answered all of my minor comments satisfactorily.

I was also asked to comment on the responses to reviewer 2:

Reviewer 2 raises the point of novelty regarding the MacRae et al paper published in 2023. I tend to disagree with the reviewer on this point because AlphaFold (although having many groundbreaking advantages) often predicts resting-state (low energy) structures which may not be meaningful to the reaction mechanism. There is a substantial difference between predicting a structure and the achievement of Yeow et al in solving a high-resolution CryoEM structure of a membrane protein complex within a membrane disc. Authors answer this question satisfactorily.

Reviewer 2 raised a good point regarding the functionality of the salt-bridges between A and C. The Authors added additional data to support the conclusion that these interactions are important for function.

Reviewer 2 suggests that a solved structure that has been non-specifically crosslinked (GraFix) would have been better. I disagree on this point. Artificial intra-complex interactions could be observed with such techniques. Notwithstanding, such a set of experiments would constitute a separate stand-alone paper.

Finally, given that functional changes have not been observed for the K184C substitution, Yeow et al have satisfactorily addressed this concern.

Point-by-point responses to reviewer comments
(Reviewer comments in black, responses in red)

REVIEWERS' COMMENTS

Reviewer #1 (Remarks to the Author):

Yeow and co-workers, have answered all of my minor comments satisfactorily.

We thank Reviewer #1 for the thorough review of this manuscript, the encouraging comments, and valuable feedback.

I was also asked to comment on the responses to reviewer 2:

Reviewer 2 raises the point of novelty regarding the MacRae et al paper published in 2023. I tend to disagree with the reviewer on this point because AlphaFold (although having many groundbreaking advantages) often predicts resting-state (low energy) structures which may not be meaningful to the reaction mechanism. There is a substantial difference between predicting a structure and the achievement of Yeow et al in solving a high-resolution CryoEM structure of a membrane protein complex within a membrane disc. Authors answer this question satisfactorily.

Reviewer 2 raised a good point regarding the functionality of the salt-bridges between A and C. The Authors added additional data to support the conclusion that these interactions are important for function.

Reviewer 2 suggests that a solved structure that has been non-specifically crosslinked (GraFix) would have been better. I disagree on this point. Artificial intra-complex interactions could be observed with such techniques. Notwithstanding, such a set of experiments would constitute a separate stand-alone paper.

Finally, given that functional changes have not been observed for the K184C substitution, Yeow et al have satisfactorily addressed this concern.

We also thank Reviewer #1 for the positive comments and feedback on our reply to Reviewer #2.